# LARGE SCALE MULTI-DOMAIN MULTI-TASK LEARNING WITH MULTIMODEL

## ABSTRACT

Deep learning yields great results across many fields, from speech recognition, image classification, to translation. But for each problem, getting a deep model to work well involves research into the architecture and a long period of tuning. We present a single model that yields good results on a number of problems spanning multiple domains. In particular, this single model is trained concurrently on ImageNet, multiple translation tasks, image captioning (COCO dataset), a speech recognition corpus, and an English parsing task. Our model architecture incorporates building blocks from multiple domains. It contains convolutional layers, an attention mechanism, and sparsely-gated layers. Each of these computational blocks is crucial for a subset of the tasks we train on. Interestingly, even if a block is not crucial for a task, we observe that adding it never hurts performance and in most cases improves it on all tasks. We also show that tasks with less data benefit largely from joint training with other tasks, while performance on large tasks degrades only slightly if at all.

## 1 INTRODUCTION

Recent successes of deep neural networks have spanned many domains, from computer vision (Krizhevsky et al., 2012) to speech recognition (Dahl et al., 2012) and many other tasks. Convolutional networks excel at tasks related to vision, while recurrent neural networks have proven successful at natural language processing tasks, e.g., at machine translation (Sutskever et al., 2014; Bahdanau et al., 2014; Cho et al., 2014). But in each case, the network was designed and tuned specifically for the problem at hand. This limits the impact of deep learning, as this effort needs to be repeated for each new task. It is also very different from the general nature of the human brain, which is able to learn many different tasks and benefit from transfer learning. The natural question arises:

*Can we create a unified deep learning model to solve tasks across multiple domains?*

The question about multi-task models has been studied in many papers in the deep learning literature. Natural language processing models have been shown to benefit from a multi-task approach a long time ago (Collobert & Weston, 2008), and recently multi-task machine translation models (Minh-Thang Luong, 2015) have even been shown to exhibit zero-shot learning when trained on multiple languages (Melvin Johnson, 2016). Speech recognition has also been shown to benefit from multi-task training (Seltzer & Droppo, 2013), as have some vision problems, such as facial landmark detection (Zhang Z., 2014). But all these models are trained on other tasks *from the same domain*: translation tasks are trained with other translation tasks, vision tasks with other vision tasks, speech tasks with other speech tasks. Multi-modal learning has been shown to improve learned representations in the unsupervised setting (Ngiam et al., 2011) and when used as a-priori known unrelated tasks (Romera-Paredes et al., 2012). But no competitive multi-task multi-modal model has been proposed, so the above question remains unanswered.

In this work, we take a step toward positively answering the above question by introducing the *MultiModel* architecture, a single deep-learning model that can simultaneously learn multiple tasks from various domains. Concretely, we train the MultiModel simultaneously on the following 8 corpora:

---

Code available at `redacted`.

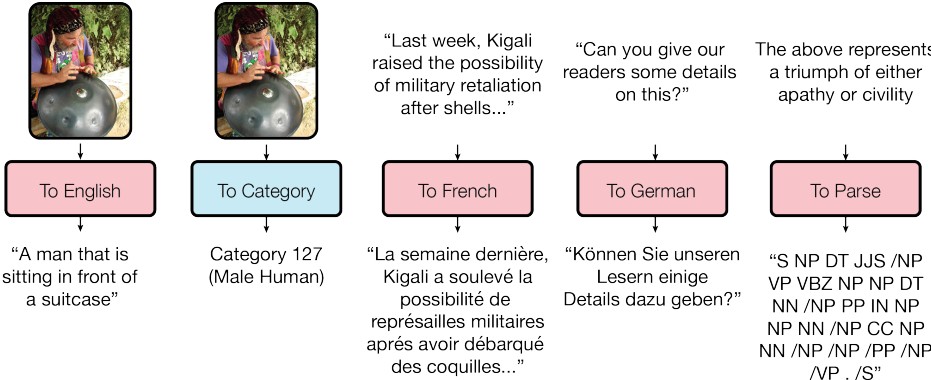

Figure 1: Examples decoded from a single MultiModel trained jointly on 8 tasks. Red depicts a language modality while blue depicts a categorical modality.

(1) WSJ speech corpus (Consortium et al., 1994), used for sentence-level speech recognition.

(2) ImageNet dataset (Russakovsky et al., 2015), used for image classification.

(3) COCO image captioning dataset (Lin et al., 2014), used for image captioning.

(4) WSJ parsing dataset (Marcus et al., 1999), used for constituency parsing.

(5) WMT English-German translation corpus, used for translation.

(6) The reverse of the above: German-English translation.

(7) WMT English-French translation corpus.

(8) The reverse of the above: French-English translation.

These corpora were chosen as they are commonly used for machine learning the respective tasks: speech-to-text, image classification, captioning, parsing and translation. The model learns all of these tasks and achieves good performance: not state-of-the-art at present, but above many task-specific models studied in recent past (see the Section 3 for details). Figure 1 illustrates some decodes taken directly from the model: it is clear that it can caption images, categorize them, translate to French and German and construct parse trees. While the MultiModel is only a first step and will be improved in the future, two key insights are crucial to making it work at all and are our main contributions.

**Small modality-specific sub-networks convert into a unified representation and back from it.** To allow training on input data of widely different sizes and dimensions, such as images, sound waves and text, we need sub-networks to convert inputs into a joint representation space. We call these sub-networks *modality nets* as they are specific to each modality (images, speech, text) and define transformations between these external domains and a unified representation. We design modality nets to be computationally minimal, promoting heavy feature extraction and ensuring that the majority of computation is performed within the domain-agnostic body of the model. Since our model is auto-regressive, modality nets need to both convert the inputs into the unified representation and later convert from this representation into the output space. Two design decisions were important:

- *The unified representation is variable-size.* While a fixed-size representation is tempting and easier to implement, it creates a bottleneck and limits the performance of the model.

- *Different tasks from the same domain share modality nets.* We avoid creating a sub-network for every task, and prefer only to create one for every input modality. For example, all translation tasks share the same modality-net (and vocabulary), no matter for which language pair. This encourages generalization across tasks and allows to add new tasks on the fly.

**Computational blocks of different kinds are crucial for good results on various problems.** The body of the MultiModel incorporates building blocks from mutiple domains. We use depthwise-separable convolutions, an attention mechanism, and sparsely-gated mixture-of-experts layers. These blocks were introduced in papers that belonged to different domains and were not studied before on tasks from other domains. For example, separable convolutions were introduced in the Xception

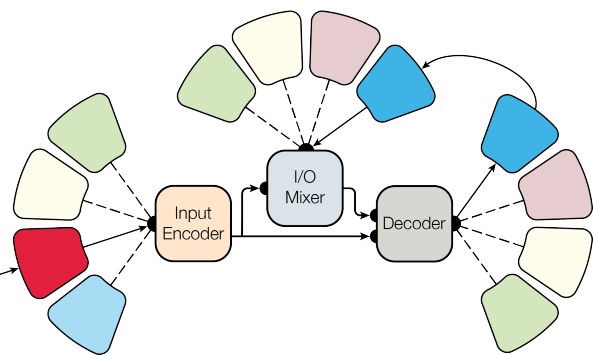

Figure 2: The MultiModel, with modality-nets, an encoder, and an autoregressive decoder.

architecture (Chollet, 2016) and were not applied to text or speech processing before. On the other hand, the sparsely-gated mixture-of-experts (Shazeer et al., 2017) had been introduced for language processing tasks and has not been studied on image problems. We find that each of these mechanisms is indeed crucial for the domain it was introduced, e.g., attention is far more important for language-related tasks than for image-related ones. But, interestingly, adding these computational blocks never hurts performance, even on tasks they were not designed for. In fact we find that both attention and mixture-of-experts layers slightly improve performance of MultiModel on ImageNet, the task that needs them the least.

## 2 MULTIMODEL ARCHITECTURE

The MultiModel consists of a few small modality-nets, an encoder, I/O mixer, and an autoregressive decoder, as depicted in Figure 2. As already said above, the encoder and decoder are constructed using 3 key computational blocks to get good performance across different problems:

(1) Convolutions allow the model to detect local patterns and generalize across space.

(2) Attention layers allow to focus on specific elements to improve performance of the model.

(3) Sparsely-gated mixture-of-experts gives the model capacity without excessive computation cost.

We start by describing the architecture of each of these 3 blocks and then introduce the encoder, decoder and the architecture of our modality-nets.

### 2.1 CONVOLUTIONAL BLOCKS

To perform local computation, we use blocks of convolutions with ReLU non-linearities and normalization. A block of convolutions gets as input a tensor of shape [batch size, sequence length, feature channels] and returns a tensor of the same shape, processed as follows.

For convolution operations, we use depthwise separable convolutions, studied for images in (Chollet, 2016), in a way similar to (Kaiser et al., 2017). Depthwise separable convolutions are a parameter- and computationally-efficient variant of the traditional convolution. They are defined by a convolution on each feature channel separately, followed by a pointwise convolution to project to the desired feature depth. We refer the reader to (Chollet, 2016) for a complete definition; here we will denote a depthwise separable convolution with weights $W^{h \times w}$ corresponding to $f$ kernels of size $h \times w$ applied to an input tensor $x$ with stride $s$ and dilated by a factor $d$ (see (Yu & Koltun, 2015)) as $SepConv_{d,s,f}(W, x)$. Note that subscripts for stride, dilation and output size are omitted when dilation $d$ or stride $s$ are equal to 1, or output size $f$ is equal to the input's feature depth.

We use convolutions in blocks that consist of three components: a $ReLU$ activation of the inputs, followed by a $SepConv$, followed by layer normalization. Layer normalization (Ba et al., 2016) acts over the $h$ hidden units of the layer below, computing layer-wise statistics for each batch example and normalizing accordingly. These normalized units are then scaled and shifted by scalar learned

parameters $G$ and $B$ respectively, producing the final units to be activated by a non-linearity. The complete convolution step is therefore defined as:

$$ConvStep_{d,s,f}(W,x) = LN(SepConv_{d,s,f}(W, ReLU(x))).$$

The convolutional steps are composed into blocks by stacking them and adding residual connections (He et al., 2016) as depicted in Figure 3. We use stacks of four convolutional blocks with two skip-connections between the stack input and the outputs of the second and fourth convolutional steps, and with the first two having $3 \times 1$ kernels and the next two having $15 \times 1$ kernels, with the final one dilated by $8$ to provide a wide receptive field. We also add $40\%$ dropout at the end of each block, so the complete block is defined as follows:

$$hidden1(x) = ConvStep(W_{h1}^{3\times1}, x)$$
$$hidden2(x) = x + ConvStep(W_{h2}^{3\times1}, hidden1(x))$$
$$hidden3(x) = ConvStep(W_{h3}^{15\times1}, hidden2(x))$$
$$hidden4(x) = x + ConvStep_{d=8}(W_{h4}^{15\times1}, hidden3(x))$$
$$ConvBlock(x) = \begin{cases} Dropout(hidden4(x), 0.4) & \text{during training} \\ hidden4(x) & \text{otherwise} \end{cases}$$

## 2.2 ATTENTION BLOCKS

For attention, we use a multi-head dot-product attention mechanism inspired by (Bahdanau et al., 2014) and similar to (Ashish Vaswani, 2017), as depicted in Figure 3. The inputs to the attention layer are two tensors: a $source$ tensor and a $target$ tensor both with the shape [batch size, sequence length, feature channels] The $target$ tensor is additively composed with a timing signal and mixed using two convolutional blocks. This mixed tensor is then self-attended using a multi-head dot-product attention, which is a dot-product attention with inputs split into $g = 8$ separate tensors representing each attention head, as shown in Figure 3. The timing signals are the main difference between this attention mechanism and the ones used previously. They allow this content-based attention to focus based on their position. They are constructed by concatenating sine and cosine curves:

$$\Delta(2d) = 1e4^{-\frac{2d}{depth}}$$
$$timing(t, [2d, 2d+1]) = [\sin(t\Delta(2d)) \parallel_2 \cos(t\Delta(2d))]$$

where $[a\|_d b]$ represent concatenation of $a$ and $b$ along the $d^{\text{th}}$ dimension. The $source$ tensor is finally passed through two different pointwise convolutions to generate the memory keys $K$ and values $V$ and the query keys, memory keys and memory values are used to apply the attention mechanism between the self-attended $target$ and the $source$ (see Figure 3).

## 2.3 MIXTURE-OF-EXPERTS BLOCKS

We use sparsely-gated mixture-of-experts layers of the same kind as introduced in (Shazeer et al., 2017): A mixture-of-experts layer consists of a number of simple feed-forward neural networks (experts) and a trainable gating network which selects a sparse combination of the experts to process each input. We refer the reader to (Shazeer et al., 2017) for details as we use exactly the architecture described there. In particular, during training we select $k = 4$ experts out of the whole expert pool and add the additional load-balancing cost as in (Shazeer et al., 2017). In each of the two mixture-of-experts layers in our model, we use a pool of $240$ experts when training on 8 problems jointly, and $60$ experts when training on each problem separately.

## 2.4 ENCODER AND MIXER AND DECODER

The body of the MultiModel consists of 3 parts: the encoder that only processes the inputs, the mixer that mixes the encoded inputs with previous outputs (autoregressive part), and a decoder that processes the inputs and the mixture to generate new outputs.

The encoder, mixer and decoder are structured similarly to previous fully convolutional sequence to sequence models such as ByteNet (Kalchbrenner et al., 2016) or WaveNet (van den Oord et al.,

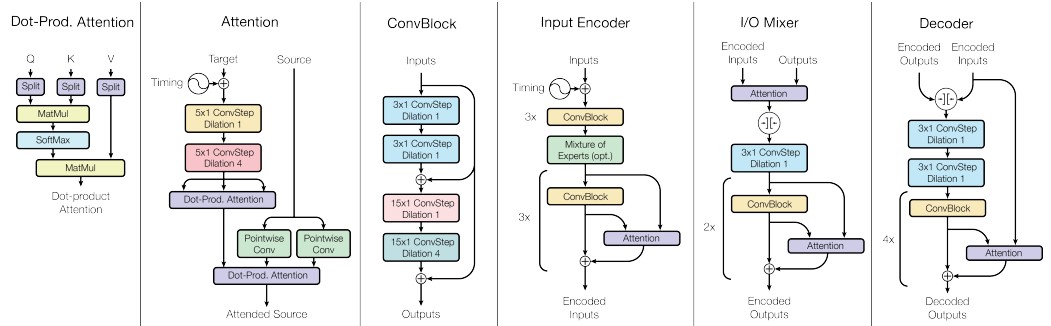

Figure 3: Architecture of the MultiModel; see text for details.

2016), but differ in the computational blocks that are used. We depict their architecture in Figure 3. As can be seen there, the encoder consists of 6 repeated convolutional blocks (described before) with a mixture-of-experts layer in the middle. The mixer consists of an attention block and 2 convolutional blocks. The decoder consists of 4 blocks of convolutions and attention, with a mixture-of-experts layer in the middle. Crucially, the convolutions in the mixer and decoder are padded on the left, so they can never access any information in the future. This allows the model to be autoregressive, and this convolutional autoregressive generation scheme offers large receptive fields over the inputs and past outputs, which are capable of establishing long term dependencies.

To allow the decoder to produce outputs for different tasks even with the same modality, we always start decoding with a command-token, such as *To-English* or *To-Parse-Tree*. We learn an embedding vector corresponding to each of the tokens during training.

## 2.5 MODALITY NETS

We have 4 modality nets, for language (text data), images, audio, and categorical data. For all predictions, we use the cross-entropy loss, per subword-unit on text, per category on classification.

### 2.5.1 LANGUAGE MODALITY NET

Our language-based data is all tokenized using the same vocabulary with 8k subword-units, following the method from (Sennrich et al., 2015). The language input modality takes a sequence of tokens ending in a termination token. This sequence of tokens is mapped to the correct dimensionality for the body using a learned embedding. On the output side, the language modality takes the decoded output of the body and performs a learned linear mapping, followed by a $Softmax$, resulting in a probability distribution over the token vocabulary.

$$LanguageModality_{\text{in}}(x, W_E) = W_E \cdot x$$
$$LanguageModality_{\text{out}}(x, W_S) = Softmax(W_S \cdot x)$$

### 2.5.2 IMAGE MODALITY NET

The image input modality is analogous to the Xception entry flow (Chollet, 2016). The input image's feature depth is gradually deepened using residual convolution blocks which we call $ConvRes$ and define as follows:

$$c1(x, F) = ConvStep_{f=F}(W^{3\times3}, x)$$
$$c2(x, F) = ConvStep_{f=F}(W^{3\times3}, c1(x, F))$$
$$p1(x, F) = MaxPool_2([3 \times 3], c2(x, F))$$
$$ConvRes(x, F) = p1(x, F) + ConvStep_{s=2}(W^{1\times1}, x),$$

where $MaxPool_s([h \times w], x)$ is a max-pooling layer over $x$ with stride $s$ and window shape $[h \times w]$. The ImageModality input flow with network depth $d$ (we use $d = 1024$) is defined as:

$$h1(x) = ConvStep_{s=2,f=32}(W^{3 \times 3}, x)$$
$$h2(x) = ConvStep_{f=64}(W^{3 \times 3}, h1(x))$$
$$r1(x) = ConvRes(h2(x), 128)$$
$$r2(x) = ConvRes(r1(x), 256)$$
$$ImageModality_{in}(x) = ConvRes(r2(x), d)$$

### 2.5.3 CATEGORICAL MODALITY NET

The categorical output modality is analogous to the Xception exit flow (Chollet, 2016). If the network inputs are two-dimensional data such as image or spectral audio data, then the one-dimensional output from the model body is first reshaped into two-dimensions again, followed by progressive down-sampling:

$$skip(x) = ConvStep_{s=2}(W_{skip}^{3 \times 3}, x)$$
$$h1(x) = ConvStep(W_{h1}^{3 \times 3}, x)$$
$$h2(x) = ConvStep(W_{h2}^{3 \times 3}, h1(x))$$
$$h3(x) = skip(x) + MaxPool_2([3 \times 3], h2(x))$$
$$h4(x) = ConvStep_{f=1536}(W_{h4}^{3 \times 3}, h3(x))$$
$$h5(x) = ConvStep_{f=2048}(W^{3 \times 3}, h4(x))$$
$$h6(x) = GlobalAvgPool(ReLU(h5(x)))$$
$$CategoricalModality_{out}(x) = PointwiseConv(W^{classes}, h6(x))$$

$GlobalAvgPool$ denotes a mean taken across all spatial and temporal dimensions.

### 2.5.4 AUDIO MODALITY NET

We accept audio input in the form of a 1-dimensional waveform over time (Golik et al., 2015; Palaz et al., 2015; Sainath et al., 2015) or as a 2-dimensional spectrogram. Both the waveform and spectral input modalities use a stack of 8 $ConvRes$ blocks from the $ImageInputModality$ (Section 2.5.2). The $i$th block has the form: $l_i = ConvRes(l_{i-1}, 2^i)$. The spectral modality does not perform any striding along the frequency bin dimension, preserving full resolution in the spectral domain.

### 2.6 EXPERIMENTAL SETUP

The modalities of the MultiModel allows to perform a training step on a batch of data from any of the 8 tasks we consider. For example, when making a training step on a batch of translation data, only the language modality sub-network will be activated. Training will then update the parameters of the language modality and all shared parameters, i.e., those in input encoder, mixer and decoder.

MultiModel can be trained on a single machine, but we used distributed training for the multi-task runs. When training jointly on 8 tasks, we had a separate worker training on each task, while the shared parameters of the model were on a parameter server and were updated asynchronously. When training on a single task, we used only a single worker training for a similar number of steps.

In all training runs report below we used the same set of hyper-parameters and the Adam optimizer (Kingma & Ba, 2014) with gradient clipping. We will release the implementation as open-source together with the details of our setup and all used hyper-parameters.

### 2.7 RELATED MODELS

The MultiModel architecture draws from earier encoder-decoder architectures applied to neural machine translation. Earlier sequence-to-sequence models for translation (Sutskever et al., 2014; Bahdanau et al., 2014; Cho et al., 2014) used recurrent neural networks (RNNs) with long short-term

| Problem | MultiModel (joint 8-problem) | State of the art |
|---|---|---|
| ImageNet (top-5 accuracy) | 86% | 95% |
| WMT EN $\rightarrow$ DE (BLEU) | 21.2 | 28.4 |
| WMT EN $\rightarrow$ FR (BLEU) | 30.5 | 41.0 |

Table 1: Comparing MultiModel to state-of-the-art (Szegedy et al., 2016; Vaswani et al., 2017).

memory cells (Hochreiter & Schmidhuber, 1997)). Convolutional architectures yielded good results on word-level neural machine translation starting from (Kalchbrenner & Blunsom, 2013) and later in (Meng et al., 2015). These early models used a standard RNN on top of the convolution to generate the output and had a bottleneck there that hurt performance, especially on longer sentences, similarly to the limitations of RNN sequence-to-sequence models without attention (Sutskever et al., 2014; Cho et al., 2014). Fully convolutional neural machine translation without this bottleneck was presented in (Kaiser & Bengio, 2016; Kalchbrenner et al., 2016). The model in (Kaiser & Bengio, 2016) (Extended Neural GPU) used a recurrent stack of gated convolutional layers, while the model in (Kalchbrenner et al., 2016) (ByteNet) did away with recursion and used left-padded convolutions in the decoder. This idea, introduced in WaveNet (van den Oord et al., 2016) and also used in MultiModel (see above) significantly improves efficiency. Depthwise separable convolutions were first studied by Sifre (Sifre & Mallat, 2013) and later they were used to get good results on large-scale image classification with Xception (Chollet, 2016).

## 3 EXPERIMENTS

We implemented the MultiModel architecture described above using TensorFlow and trained it in a number of configurations. We focused our experiments so as to answer the following questions:

(1) How far is the MultiModel trained on 8 tasks simultaneously from state-of-the-art results?

(2) How does training on 8 tasks simultaneously compare to training on each task separately?

(3) How do the different computational blocks discussed above influence different tasks?

In answering the above questions, we don't always consider all 8 problems. Especially the 4 translation problems behave very similarly, so we decided to not include them all in each comparison but we focused on the more varied problems instead.

To answer question (1), we compare the performance of the 8-problem MultiModel with state-of-the-art results in Table 1. We use the standard top-5 accuracy metric for ImageNet and the standard BLEU metric for translation (scored with MOSES on newstest2014 while newstest2013 was used as the development set). We did not invest much time yet in tuning hyper-parameters of the MultiModel, so we believe that the difference seen there will become much smaller with more tuning. The results we achieve are similar to the ones task-specific models get without heavy tuning, e.g., on English-French translation we improve on the recent Extended Neural GPU results (Kaiser & Bengio, 2016).

To answer question (2), we compare the MultiModel trained jointly with MultiModel trained separately just on a single task. Since we are comparing different instantiations of the same model, we report two internal metrics: the negative log-perplexity and per-token accuracy (measured on the development set). As can be seen from the results in Table 2, the joint 8-problem model performs similarly to single-model on large tasks, and better, sometimes significantly, on tasks where less data is available, such as parsing.

The large improvement on parsing seen in Table 2 is not that surprising taking into account the large number of text data in translation tasks. But we were curious if training parsing just with ImageNet, a seemingly unrelated task, would also bring any improvements. This is indeed the case, as can be seen in Table 3. The difference in performance is significant, and since we use both dropout and early stopping, we conjecture that it is not related to over-fitting. Rather, it seems, there are computational primitives shared between different tasks that allow for some transfer learning even between such seemingly unrelated tasks as ImageNet and parsing.

| Problem | Joint 8-problem | | Single problem | |
|---|---|---|---|---|
| | log(perpexity) | accuracy | log(perplexity) | accuracy |
| ImageNet | 1.7 | 66% | 1.6 | 67% |
| WMT EN→DE | 1.4 | 72% | 1.4 | 71% |
| WSJ speech | 4.4 | 41% | 5.7 | 23% |
| Parsing | 0.15 | 98% | 0.2 | 97% |

Table 2: Comparison of the MultiModel trained jointly on 8 tasks and separately on each task.

| Problem | Alone | | | W/ ImageNet | | | W/ 8 Problems | | |
|---|---|---|---|---|---|---|---|---|---|
| | log(ppl) | acc. | full | log(ppl) | acc. | full | log(ppl) | acc. | full |
| Parsing | 0.20 | 97.1% | 11.7% | 0.16 | 97.5% | 12.7% | 0.15 | 97.9% | 14.5% |

Table 3: Results on training parsing alone, with ImageNet, and with 8 other tasks. We report log-perplexity, per-token accuracy, and the percentage of fully correct parse trees.

To answer question (3), we check how training without the mixture-of-experts layers or without the attention mechanism influences performance on different problems. Since both these mechanisms were designed with machine translation in mind, we check the English-French translation. But we also include ImageNet, since this is the problem that stands the least to benefit from those blocks. In fact, one could expect that removing these blocks will improve performance on ImageNet alone if they were truly useless for this task. In contrast, we see in Table 4 that these blocks either don't affect or slightly improve performance. This leads us to conclude that mixing different computation blocks is in fact a good way to improve performance on many various tasks.

## 4 CONCLUSIONS

We demonstrate, for the first time, that a single deep learning model can jointly learn a number of large-scale tasks from multiple domains. The key to success comes from designing a multi-modal architecture in which as many parameters as possible are shared and from using computational blocks from different domains together. We believe that this treads a path towards interesting future work on more general deep learning architectures, especially since our model shows transfer learning from tasks with a large amount of available data to ones where the data is limited.

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

| Problem | All Blocks | | Without MoE | | Without Attention | |
|---|---|---|---|---|---|---|
| | log(perpexity) | accuracy | log(perplexity) | accuracy | log(perplexity) | accuracy |
| ImageNet | 1.6 | 67% | 1.6 | 66% | 1.6 | 67% |
| WMT EN→FR | 1.2 | 76% | 1.3 | 74% | 1.4 | 72% |

Table 4: Ablating mixture-of-experts and attention from MultiModel training.

Dzmitry Bahdanau, Kyunghyun Cho, and Yoshua Bengio. Neural machine translation by jointly learning to align and translate. *CoRR*, abs/1409.0473, 2014. URL http://arxiv.org/abs/1409.0473.

Kyunghyun Cho, Bart van Merrienboer, Caglar Gulcehre, Fethi Bougares, Holger Schwenk, and Yoshua Bengio. Learning phrase representations using rnn encoder-decoder for statistical machine translation. *CoRR*, abs/1406.1078, 2014. URL http://arxiv.org/abs/1406.1078.

Francois Chollet. Xception: Deep learning with depthwise separable convolutions. *arXiv preprint arXiv:1610.02357*, 2016.

Ronan Collobert and Jason Weston. A unified architecture for natural language processing: deep neural networks with multitask learning. In *Proceedings of the 25th International Conference on Machine learning*, pp. 160–167, 2008.

Linguistic Data Consortium et al. Csr-ii (wsj1) complete. *Linguistic Data Consortium, Philadelphia, vol. LDC94S13A*, 1994.

George E. Dahl, Dong Yu, Li Deng, and Alex Acero. Context-dependent pre-trained deep neural networks for large-vocabulary speech recognition. *IEEE Transactions on Audio, Speech & Language Processing*, 20(1):30–42, 2012.

Pavel Golik, Zoltán Tüske, Ralf Schlüter, and Hermann Ney. Convolutional neural networks for acoustic modeling of raw time signal in LVCSR. In *Proc. of INTERSPEECH'15*, pp. 26–30, 2015.

Kaiming He, Xiangyu Zhang, Shaoqing Ren, and Jian Sun. Deep residual learning for image recognition. In *2016 IEEE Conference on Computer Vision and Pattern Recognition, CVPR'16*, pp. 770–778, 2016.

Sepp Hochreiter and Jürgen Schmidhuber. Long short-term memory. *Neural computation*, 9(8): 1735–1780, 1997.

Łukasz Kaiser and Samy Bengio. Can active memory replace attention? In *Advances in Neural Information Processing Systems, (NIPS)*, 2016.

Łukasz Kaiser, Aidan N. Gomez, and Francois Chollet. Depthwise separable convolutions for neural machine translation. *arXiv preprint arXiv:1706.03059*, 2017.

Nal Kalchbrenner and Phil Blunsom. Recurrent continuous translation models. In *Proceedings EMNLP 2013*, pp. 1700–1709, 2013. URL http://nal.co/papers/KalchbrennerBlunsom_EMNLP13.

Nal Kalchbrenner, Lasse Espeholt, Karen Simonyan, Aaron van den Oord, Alex Graves, and Koray Kavukcuoglu. Neural machine translation in linear time. *arXiv preprint arXiv:1610.10099*, 2016.

Diederik P. Kingma and Jimmy Ba. Adam: A method for stochastic optimization. *CoRR*, abs/1412.6980, 2014. URL http://arxiv.org/abs/1412.6980.

Alex Krizhevsky, Ilya Sutskever, and Geoffrey Hinton. Imagenet classification with deep convolutional neural network. In *Advances in Neural Information Processing Systems*, 2012.

Tsung-Yi Lin, Michael Maire, Serge J. Belongie, Lubomir D. Bourdev, Ross B. Girshick, James Hays, Pietro Perona, Deva Ramanan, Piotr Dollár, and C. Lawrence Zitnick. Microsoft COCO: common objects in context. *CoRR*, abs/1405.0312, 2014. URL http://arxiv.org/abs/1405.0312.

Mitchell P Marcus, Beatrice Santorini, Mary Ann Marcinkiewicz, and Ann Taylor. Treebank-3 ldc99t42. *CD-ROM. Philadelphia, Penn.: Linguistic Data Consortium*, 1999.

Quoc V. Le Maxim Krikun Yonghui Wu Zhifeng Chen Nikhil Thorat Fernanda Viégas Martin Wattenberg Greg Corrado Macduff Hughes Jeffrey Dean Melvin Johnson, Mike Schuster. Google's multilingual neural machine translation system: Enabling zero-shot translation. *arXiv preprint arXiv:1611.04558*, 2016.

Fandong Meng, Zhengdong Lu, Mingxuan Wang, Hang Li, Wenbin Jiang, and Qun Liu. Encoding source language with convolutional neural network for machine translation. In *ACL*, pp. 20–30, 2015.

Ilya Sutskever Oriol Vinyals Łukasz Kaiser Minh-Thang Luong, Quoc V. Le. Multi-task sequence to sequence learning. *arXiv preprint arXiv:1511.06114*, 2015.

Jiquan Ngiam, Aditya Khosla, Mingyu Kim, Juhan Nam, Honglak Lee, and Andrew Y. Ng. Multi-modal deep learning. In *Proceedings of ICML'11*, pp. 689–696, 2011.

Dimitri Palaz, Mathew Magimai-Doss, and Ronan Collobert. Convolutional neural networks-based continuous speech recognition using raw speech signal. In *Proc. of ICASSP'15*, pp. 4295–4299, 2015.

Bernardino Romera-Paredes, Andreas Argyriou, Nadia Berthouze, and Massimiliano Pontil. Exploiting unrelated tasks in multi-task learning. In *JMLR Proceedings of AISTATS'12*, pp. 951–959, 2012.

Olga Russakovsky, Jia Deng, Hao Su, Jonathan Krause, Sanjeev Satheesh, Sean Ma, Zhiheng Huang, Andrej Karpathy, Aditya Khosla, Michael Bernstein, Alexander C. Berg, and Li Fei-Fei. ImageNet Large Scale Visual Recognition Challenge. *International Journal of Computer Vision (IJCV)*, 115 (3):211–252, 2015. doi: 10.1007/s11263-015-0816-y.

Tara N. Sainath, Ron J. Weiss, Andrew W. Senior, Kevin W. Wilson, and Oriol Vinyals. Learning the speech front-end with raw waveform cldnns. In *Proc. of INTERSPEECH'15*, pp. 1–5, 2015.

Michael L. Seltzer and Jasha Droppo. Multi-task learning in deep neural networks for improved phoneme recognition. In *Proceedings of the IEEE International Conference on Acoustics, Speech and Signal Processing (ICASSP'13)*, 2013.

Rico Sennrich, Barry Haddow, and Alexandra Birch. Neural machine translation of rare words with subword units. *CoRR*, 2015.

Noam Shazeer, Azalia Mirhoseini, Krzysztof Maziarz, Andy Davis, Quoc Le, Geoffrey Hinton, and Jeff Dean. Outrageously large neural networks: The sparsely-gated mixture-of-experts layer. *arXiv preprint 1701.06538*, 2017.

Laurent Sifre and Stéphane Mallat. Rotation, scaling and deformation invariant scattering for texture discrimination. In *2013 IEEE Conference on Computer Vision and Pattern Recognition, Portland, OR, USA, June 23-28, 2013*, pp. 1233–1240, 2013.

Ilya Sutskever, Oriol Vinyals, and Quoc VV Le. Sequence to sequence learning with neural networks. In *Advances in Neural Information Processing Systems*, pp. 3104–3112, 2014. URL http://arxiv.org/abs/1409.3215.

Christian Szegedy, Sergey Ioffe, and Vincent Vanhoucke. Inception-v4, inception-resnet and the impact of residual connections on learning. *CoRR*, abs/1602.07261, 2016.

Aäron van den Oord, Sander Dieleman, Heiga Zen, Karen Simonyan, Oriol Vinyals, Alex Graves, Nal Kalchbrenner, Andrew Senior, and Koray Kavukcuoglu. Wavenet: A generative model for raw audio. *CoRR abs/1609.03499*, 2016.

Ashish Vaswani, Noam Shazeer, Niki Parmar, Jakob Uszkoreit, Llion Jones, Aidan N. Gomez, Lukasz Kaiser, and Illia Polosukhin. Attention is all you need. *CoRR*, 2017. URL http://arxiv.org/abs/1706.03762.

Fisher Yu and Vladlen Koltun. Multi-scale context aggregation by dilated convolutions. *arXiv preprint arXiv:1511.07122*, 2015.

Loy C.C. Tang X. Zhang Z., Luo P. Facial landmark detection by deep multi-task learning. In *Proceedings of ECCV'14*, 2014.

