# OpenReview forum: "Large Scale Multi-Domain Multi-Task Learning with MultiModel"
_ICLR.cc/2018/Conference — Reject_

### Official Review · AnonReviewer2 · 2017-11-21
**Deep architecture, shallow insights**

**Rating:** 3
**Confidence:** 5

**Review:**

The paper describes a neural end-to-end architecture to solve multiple tasks at once.  The architecture consists of an encoder, a mixer, a decoder, and many modality networks to cover different types of input and output pairs for different tasks.  The engineering endeavor is impressive, but the paper has little scientific value.  Below are a few suggestions to make the paper stronger.

It is possible that the encoder, mixer, and decoder are just multiplexing tasks based on the input.  One way to analyze whether this happens is to predict the identity of the task from the hidden vectors.  If this is the case, how to prevent it from happening?  If this does not happen, what is being shared across tasks?  This can be analyzed by embedding the inputs from different tasks and looking for inputs from other tasks within a neighborhood in the embedding space.

Why multitask learning help the model perform better is still unclear.  If the model is able to leverage knowledge learned from one task to perform another task, then we expect to see either faster convergence or good performance with fewer samples.  The authors should analyze if this is the case, and if not, what are we actually benefiting from multitask learning?

If the modality network is shared across multiple tasks, we expect the learned hidden representation produced by the modality network is more universal.  If that is the case, what information of the input is being retained when training with multiple tasks and what information of the input is being discarded when training with a single task?

Reporting per-token accuracies, such as those in Table 2, is problematic.  It's unclear how to compute per-token accuracies for structured prediction tasks, such as speech recognition, parsing, and translation.  Furthermore, based on the results in Table 2, the model clearly fails on the speech recognition task.  The author should also report the standard speech recognition metric, word error rates (WER), for the speech recognition task in Table 1.

---

> ### Author Response · Authors · 2018-01-05
> **Grateful for improvement suggestions, insisting that engineering and science are intertwined in deep learning.**
>
> We are thankful to the reviewer for suggestions on how to improve the paper, which we address below. But we need to start with the main negative point, summarized in this sentence in the review: "The engineering endeavor is impressive, but the paper has little scientific value."
>
> We are the first to admit that our paper does not provide final insights on multi-task learning: we do not know exactly why the transfer learning we observe happens and we cannot pinpoint the exact parts of the joint representation that would be responsible for it. That's why we use the words "first step" in our paper. On the other hand, we had to work  hard to make this multi-task learning work at all. All key components that we describe (small modality nets, mixing of convolutions, attention, and especially large scale sparely-gated mixture-of-experts layers) are crucial to making this work. Without any single of these components, or if they are put together in a wrong way, the multi-problem training just doesn't work: the results get far worse than in single-problem training or some tasks fail to train at all.
>
> We feel that it has been the case many times in deep learning (and in science in general) that an engineering solution was presented first, and a proper detailed analysis only followed later. In fact, one could argue that we are only beginning to understand neural network representations in general, years after they were constructed. So we'd insist that presenting an engineered system, like ours, has value for the development of science later. Also, to get back from the discussions that touch on philosophy, we need to point out that the ICLR'18 call for papers states "we take a broad view of the field" and the list of relevant topics even includes "implementation issues". We believe this means that our paper fits the conference and we hope that the reviewer will take this into account and revise the score.
>
> As for the questions in the comments.
>
> * "It is possible that the encoder, mixer, and decoder are just multiplexing tasks?" Iit is not possible that they are doing just that, because we see improvements in final scores due to transfer learning. If they were only multiplexing, then there would be no transfer learning. But it is possible that they are multiplexing to a certain extent, and we are planning a future work extending the MultiModel with a domain-adversarial loss, to check if that makes transfer learning more pronounced.
>
> * "We expect to see either faster convergence or good performance with fewer samples." We do see transfer learning, e.g. on parsing, where there are fewer data samples available we see better error, going down from ~3% to ~2% on accuracy (which is correlated with other metrics, see below).
>
> * "What information of the input is being retained when training with multiple tasks and what information of the input is being discarded when training with a single task?" We believe this is a great question, but it has not even been truly answered for much simpler single-task deep networks. We believe that papers like (https://arxiv.org/abs/1703.00810) start to provide methods for answering these questions, but these are too recent for us to be able to apply to our large model at this point.
>
> * "In Table 2, it's unclear how to compute per-token accuracies for structured prediction tasks [...] the model clearly fails on the speech recognition [...] should also report WER in Table 1".
>
> We make a clear distinction in the paper between Table 1 and Table 2. In Table 2, we compare different versions of our model against each other. In that case, it is clear how to compute per-token accuracies: it's the same model operating on the same data tokenized in the same way, so we simply report the accuracy and perplexity. As has been observed many times in sequence-to-sequence models, internal per-token accuracy and perplexity correlate strongly with external metrics, so we believe that this is a good way to compare different versions of the same model (it is also the default way of model selection in neural machine translation and other sequence-to-sequence tasks). In Table 1, we try to put the results of our model in the context of other models in the field. We did not do it for all 8 tasks because there are technical difficulties, but we do get some fully correctly transcribed sentences on the dev set in speech recognition,  and in general we'd rather not report a number than report a wrong one (see the reply to Reviewer 1 above for the discussion on problems with WER). Since we are not claiming SOTA results and didn't do heavy tuning, we believe that this is a reasonable way to report our results. We are working on adding more and we will be grateful for more suggestions for the final version.
>
> We hope that the above discussion will convince the reviewer to revise the score.

---

> > ### Comment · AnonReviewer2 · 2018-01-10
> > **bring back the scaffoldings**
> >
> > Thanks for the response.
> >
> > We probably have a different definition about what science is. To me, any set of rigorous experiments with good controls is considered science. From your response, I can tell that the task that you are working on is hard, and you have made many design choices to make it work. I would consider a paper with scientific value if you document what works, what doesn't, what the potential problems are, and what individual changes make it work. That's how I interpret "implementation issues" mentioned in the call-for-papers. This is partially fulfilled in Table 3 and 4, but it is more like a post-hoc analysis, attacking a straw man rather than the real problem.
> >
> > When I ask for WERs for the speech recognition task, I'm not going to criticize you for not getting SOTA results. As long as the proposed approach has potential and merit, SOTA results are not required. When you report WERs that are comparable to other papers, it gives the readers a sense of how the model is performing. I believe that's the reason you report BLEU scores in the paper. Token error rates have very little meaning to speech researchers.
> >
> > I think we can all agree the above is the minimum requirement of a scientific study. I believe you have spent a huge effort optimizing the architecture, so you must already have all the experimental numbers to write a good paper and you could have written a paper including all these numbers and comparisons for your design decisions.
> >
> > Beyond the bare minimum, it would be great if you can give hunches and analyses about why you are having these problems, why you make certain decisions, and why those decisions work.
> >
> > Let me end my response with a quote from Carl Meyer's Matrix Analysis book.
> >
> > "Reacting to criticism concerning the lack of motivation in his writings, Gauss remarked that architects of great cathedrals do not obscure the beauty of their work by leaving the scaffolding in place after the construction has been completed. His philosophy epitomized the formal presentation and teaching of mathematics throughout the nineteenth and twentieth centuries, and it is still commonly found in mid-to-upper-level mathematics textbooks. The inherent efficiency and natural beauty of mathematics are compromised by straying too far from Gauss' viewpoint. But, as with most things in life, appreciation is generally preceded by some understanding seasoned with a bit of maturity, and in mathematics this comes from seeing some of the scaffolding."

---

> > > ### Author Response · Authors · 2018-01-12
> > > **Thanks for the analysis, please suggest more changes**
> > >
> > > We are very grateful for the suggestions on what to improve on. We will certainly add a section with more negative results on multi-task training -- we only did Table 4 and skipped ablations where nothing worked because it is unusual to report very bad results, but we will happily put it back. We will also add more intuitions about the architectural choices, is the beginning of Section 2 the right place? We would be grateful for even more suggestions and concrete points to improve in the final version of the paper.

---

> > > > ### Comment · AnonReviewer2 · 2018-01-12
> > > > **concrete suggestions**
> > > >
> > > > > we only did Table 4 and skipped ablations where nothing worked because it is unusual to report very bad results, but we will happily put it back.
> > > >
> > > > Yes, that's what I was aiming for. I'm not sure what architecture you started with, hopefully a more standard one (if not, justify). Pick a few of your tasks (ideally two). Run your base model and show the readers that if you use the naive solution you would get really bad results. It would be great if you can run the base model on the individual tasks and compare. These results demonstrate the problem we are facing, and ideally should go to the end of section 1.
> > > >
> > > > In section 2, you introduce several basic components. I believe you have some intuitions why these components are useful. I think this section is already good, but try to add more motivation to each components. For example, some of the components are more problem agnostic, and some of them are more problem specific. (This might be a good place to mention the potential multiplexing issue.) The problem specific ones are easier to justify, because they are aligned with the standard architectures for each individual tasks. You might get into trouble when explaining the problem agnostic ones, because all of these components are powerful by themselves. For example, the attention blocks and convolution blocks can pretty much do anything.
> > > >
> > > > In the experiments, first use the two-task setting in section 1, add the proposed components one by one, and show that they help. (Ideally you should analyze the why here and address the multiplexing problem. See the first post.) Repeat the experiments with different task combination and component combination. Finally report the ones with all eight tasks and show the readers the proposed architecture can indeed solve the problem introduced in section 1.
> > > >
> > > > This is how I would write the paper if I had all the results. You don't need to follow the structure, but the experimental comparisons for the minimal settings should be there. I hope these suggestions are concrete enough. Please let me know if anything is unclear and if you need more input.

---

### Official Review · AnonReviewer3 · 2017-11-23
**Multi-task learning does not hurt performance across different domains**

**Rating:** 6
**Confidence:** 4

**Review:**

The paper presents a multi-task architecture that can perform multiple tasks across multiple different domains. The authors design an architecture that works on image captioning, image classification, machine translation and parsing.

The proposed model can maintain performance of single-task models and in some cases show slight improvements. This is the main take-away from this paper.

There is a factually incorrect statement - depthwise separable convolutions were not introduced in Chollet 2016. Section 2 of the same paper also notes it (depthwise convolutions can be traced back to at least 2012).

---

> ### Author Response · Authors · 2018-01-05
> **Corrected depthwise separable statement, want to point out transfer learning.**
>
> We are grateful for pointing out the false statement about depthwise separable convolutions. We changed the statement in Section 2.1 and we still have a more complete discussion in Section 2.7 (was Section 2.6 in the previous revision). The earliest reference we could find was by Laurent Sifre and Stéphane Mallat from 2013, but we believe these were discussed even earlier. We will be grateful if the reviewer can provide us with a reference or a formulation about such earlier work and we will include it in the final version of the paper.
>
> We also want to point out that, in addition to maintaining performance of single-task models, the MultiModel shows substantial gains on tasks where only a smaller corpus is available, such as on parsing (going from ~3% per-token error to ~2% error, a large relative reduction) We find this transfer learning one of the key results of our paper and we hope that the reviewer will take it into account.

---

### Official Review · AnonReviewer1 · 2017-11-27

**Rating:** 6
**Confidence:** 3

**Review:**

The paper presents a multi-task, multi-domain model based on deep neural networks. The proposed model is able to take inputs from various domains (image, text, speech) and solves multiple tasks, such as image captioning, machine translation or speech recognition. The proposed model is composed of several features learning blocks (one for each input type) and of an encoder and an auto-regressive decoder, which are domain-agnostic. The model is evaluated on 8 different tasks and is compared with a model trained separately on each task, showing improvements on each task.

The paper is well written and easy to follow.

The contributions of the paper are novel and significant. The approach of having one model able to perform well on completely different tasks and type of input is very interesting and inspiring. The experiments clearly show the viability of the approach and give interesting insights. This is surely an important step towards more general deep learning models.

Comments:

* In the introduction where the 8 databases are presented, the tasks should also be explained clearly, as several domains are involved and the reader might not be familiar with the task linked to each database. Moreover, some databases could be used for different tasks, such as WSJ or ImageNet.

* The training procedure of the model is not explained in the paper. What is the cost function and what is the strategy to train on multiple tasks ? The paper should at least outline the strategy.

* The experiments are sufficient to demonstrate the viability of the approach, but the experimental setup is not clear. Specifically, there is an issue about the speech recognition part of the experiment. It is not clear what the task exactly is: continuous speech recognition, isolated word recognition ? The metrics used in Table 1 are also not clear, they should be explained in the text. Also, if the task is continuous speech recognition, the WER (word error rate) metric should be used. Information about the detailed setup is also lacking, specifically which test and development sets are used (the WSJ corpus has several sets).

* Using raw waveforms as audio modality is very interesting, but this approach is not standard for speech recognition, some references should be provided, such as:
P. Golik, Z. Tuske, R. Schluter, H. Ney, Convolutional Neural Networks for Acoustic Modeling of Raw Time Signal in LVCSR, in: Proceedings of the Annual Conference of the International Speech Communication Association (INTERSPEECH), 2015, pp. 26–30.
D. Palaz, M. Magimai Doss and R. Collobert, (2015, April). Convolutional neural networks-based continuous speech recognition using raw speech signal. In Acoustics, Speech and Signal Processing (ICASSP), 2015 IEEE International Conference on (pp. 4295-4299). IEEE.
T. N. Sainath, R. J. Weiss, A. Senior, K. W. Wilson, and O. Vinyals. Learning the Speech Front-end With Raw Waveform CLDNNs. Proceedings of the Annual Conference of the International Speech Communication Association (INTERSPEECH), 2015.

Revised Review:
The main idea of the paper is very interesting and the work presented is impressive. However, I tend to agree with Reviewer2, as a more comprehensive analysis should be presented to show that the network is not simply multiplexing tasks. The experiments are interesting, except for the WSJ speech task, which is almost meaningless. Indeed, it is not clear what the network has learned given the metrics presented, as the WER on WSJ should be around 5% for speech recognition.
I thus suggest to either drop the speech experiment, or the modify the network to do continuous speech recognition. A simpler speech task such as Keyword Spotting could also be investigated.

---

> ### Author Response · Authors · 2018-01-05
> **Grateful for the detailed suggestions, followed most of them in the new revision.**
>
> We'd like to thank the reviewer for the constructive suggestions. We tried to follow them all, space permitting, in the new revision of  the paper.
>
> * We added some general clarification in the list of the corpora. Specifying all details (train/dev/test splits, text normalization, tokenizers, etc.) would take a lot of space, but in the final version we will add pointers to data preparation code in our open-source code base where all these details are included.
>
> * We are very grateful for the suggestion to add an "experimental setup" section, which we did. We used cross-entropy loss for each task and trained in an asynchronous distributed way, as described now in that section.
>
> * We described the metrics used in Table 1 in the text, as suggested. We did not include WER in Table 1 for a number of technical reasons. Mainly, since we use subword tokens, to get comparable WER we need to go through de-tokenization, renormalization and re-tokenization with a comparable tokenizer, and we are not confident that this process makes for truly comparable results. Note that this a direct consequence of the multi-problem setting: we need a single tokenizer for all corpora, so we cannot afford to just use the default one for each data-set as they are different. We are now re-implementing SOTA speech-to-text models in our codebase, so by publication time we hope to get accuracy results from SOTA models, which we could use as a base for comparison. Another problem is whether we should compare to results with or without a language model: we do not use any explicit LM on top of our network, but the multi-task training can basically act as learning a language model too. In any case, we are not claiming SOTA results as we focus on transfer learning and other aspects of multi-task training.
>
> * We added the suggested references and will be grateful for more.
>
> We want to thank the reviewer again for constructive comments.

---

### Decision · Program_Chairs · 2018-01-29
**ICLR 2018 Conference Acceptance Decision**

**Decision:**

Reject

**Comment:**


Pros:
+ Interesting and promising approach to multi-domain, multi-task learning.
+ Paper is clearly written.

Cons:
- Reads more like a technical report than a research paper: more space should be devoted to explaining the design decisions behind the model and the challenges involved, as this will help others tackle similar problems.

This paper had extensive discussion between the reviewers and authors, and between the reviewers.  In the end, the reviewers want more insight into the architectural choices made, either via ablation studies or via a series of experiments in which tasks or components are added one at a time.  The consensus is that this would give readers a lot more insight into the challenges involved in tackling multiple domains and multiple tasks in a single model and a lot more guidance on how to do it.